# Systematic media review: A novel method to assess mass-trauma epidemiology in absence of databases—A pilot-study in Rwanda

**Lotta Velin**[1]*, **Mbonyintwari Donatien**[2], **Andreas Wladis**[1], **Menelas Nkeshimana**[3], **Robert Riviello**[4,5], **Jean-Marie Uwitonze**[6], **Jean-Claude Byiringiro**[2], **Faustin Ntirenganya**[2,3], **Laura Pompermaier**[1,5]

1 Department of Biomedical and Clinical Sciences, Center for Teaching & Research in Disaster Medicine and Traumatology (KMC), Linköping University, Linköping, Sweden, 2 College of Medicine and Health Sciences, University of Rwanda, Kigali, Rwanda, 3 University Teaching Hospital in Kigali, Kigali, Rwanda, 4 Brigham and Women's Hospital, Boston, MA, United States of America, 5 Department of Global Health and Social Medicine, Program in Global Surgery and Social Change, Harvard Medical School, Boston, MA, United States of America, 6 Emergency Medical Services (SAMU), Kigali, Rwanda

* lotta.velin@liu.se

**Data Availability Statement:** All relevant data are within the manuscript and its Supporting Information files. The data underlying the results

## Abstract

### Objective

Surge capacity refers to preparedness of health systems to face sudden patient inflows, such as mass-casualty incidents (MCI). To strengthen surge capacity, it is essential to understand MCI epidemiology, which is poorly studied in low- and middle-income countries lacking trauma databases. We propose a novel approach, the "systematic media review", to analyze mass-trauma epidemiology; here piloted in Rwanda.

### Methods

A systematic media review of non-academic publications of MCIs in Rwanda between January 1st, 2010, and September 1st, 2020 was conducted using NexisUni, an academic database for news, business, and legal sources previously used in sociolegal research. All articles identified by the search strategy were screened using eligibility criteria. Data were extracted in a RedCap form and analyzed using descriptive statistics.

### Findings

Of 3187 articles identified, 247 met inclusion criteria. In total, 117 MCIs were described, of which 73 (62.4%) were road-traffic accidents, 23 (19.7%) natural hazards, 20 (17.1%) acts of violence/terrorism, and 1 (0.09%) boat collision. Of Rwanda's 30 Districts, 29 were affected by mass-trauma, with the rural Western province most frequently affected. Road-traffic accidents was the leading MCI until 2017 when natural hazards became most common. The median number of injured persons per event was 11 (IQR 5–18), and median on-site deaths was 2 (IQR 1–6); with natural hazards having the highest median deaths (6 [IQR 2–18]).

presented in the study are available from Nexis Uni, which is an online database accessible through institutional agreements with universities/libraries or through direct payment for the services. The authors did not have special access to the data that others would not have. More information can be found here: https://internationalsales.lexisnexis.com/products/nexis-uni.

**Funding:** The author(s) received no specific funding for this work.

**Competing interests:** The authors have declared that no competing interests exist.

## Conclusion

In Rwanda, MCIs have decreased, although landslides/floods are increasing, preventing a decrease in trauma-related mortality. By training journalists in "mass-casualty reporting", the potential of the "systematic media review" could be further enhanced, as a way to collect MCI data in settings without databases.

## Introduction

Surge capacity is defined as "the ability of a healthcare system to expand beyond its regular operations and accommodate a greater number of patients in response to multiple casualty-producing events". [1] Mass-casualty incidents (MCIs), such as those caused by road traffic accidents or natural calamities, can lead to sudden increases of patients seeking urgent care, challenging the capacity of hospitals to deliver adequate care, particularly where the system is already weakened by restricted resources and limited workforce, as in many low- and middle-income countries (LMICs) [2].

Surge capacity is facilitated by the existence of a strong health system, but can also be further developed by designing mass-casualty protocols that guide the reorganization of health-care delivery, for example by pausing elective surgery [3], discharging patients to lower levels of care [4], calling in additional staff, or transitioning to modified triage systems [5]. However, the type of response depends on multiple factors, such as local health care organization, trauma mechanism [1], and availability of resources. Contextualized knowledge of the causes and extent of MCIs in a specific health system is necessary to develop appropriate strategies to build surge capacity.

Globally, analysis of MCI epidemiology has been restricted by the complexity of linking trauma patients to the mass-trauma event, and the lack of pre-hospital data collection about the trauma event or the number of casualties involved, hindering identification of MCIs amongst all hospital trauma cases. In settings with limited resources, challenges associated with data collection are further complicated by the fact that data is often recorded in logbooks, where incomplete or low-quality data is frequent [6]. To address these issues, "patient tracking" solutions to triage and localize patients in real-time to specific MCIs and geographical areas has been proposed, although, this has not yet been systematically implemented [7,8]. In high-income countries (HICs), surge capacity is frequently monitored and MCI protocols are developed using data collected in local databases [9–12]. However, these protocols have limited applicability in LMICs due to vast differences in preconditions, such as the disproportion of the global physician density, which is 0.14/1000 people in Rwanda, as compared to 4.25/1000 in Germany or 2.61/1000 in the U.S [13].

Rwanda is a small but densely populated low-income nation in East Africa, with hilly topography and high annual precipitations occurring primarily in two annual "rainy seasons". The Ministry of Emergency Management compiles yearly reports on disasters [14] and provides detailed national contingency plans [15–17]. However, these reports only include data on certain types of disasters, limiting its use in assessing MCIs. Since 2019, an injury registry has been implemented at the two university teaching hospitals, the military hospital and one referral hospital [18], however, it does not include data from other health facilities hindering understanding of the national epidemiology.

These obstacles led us to look for new, feasible ways to estimate the burden of MCI in limited-resource settings. This study, therefore, aims to pilot the novel methodology "systematic

media review" in Rwanda, to identify mass-trauma events and assess their epidemiology using media as the data source.

## Methods

### Defining mass-trauma/mass-casualty incidents

While "disaster" in the context of health care describes situations where available resources are not enough to provide adequate patient care [19], "mass-trauma" and "mass-casualty incidents" lack globally accepted definitions. Algorithms have been proposed to quantify the maximum number of casualties that a hospital can manage at the same time, such as the Hospital Acute Care Surge Capacity (HACSC) [20] but, have not been validated in more resource-limited contexts. In this study, we defined MCIs as events causing three or more traumatic injuries, as this was expected to be the lower threshold for when health facilities in Rwanda may need to utilize "surge capacity" to meet the patient surge.

### Search strategy

Data was collected through NexisUni [21], a database for non-academic publications, commonly used in sociological and medicolegal research [22,23]. The search strategy was developed in English with the help of an information specialist from the Harvard Medical School Countway Library (S1 Appendix) and was translated to French and Kinyarwanda. Inclusion criteria were news articles, radio/news transcripts, and governmental/non-governmental reports published between January 1st, 2010 and September 1st, 2020, which reported traumatic events that occurred in Rwanda and caused three or more injuries or, in the case of missing injury data, where the reported number of on-site deaths ≥1 and the trauma mechanism suggested the possibility of three or more injuries.

The exclusion criteria were languages other than English, French, or Kinyarwanda; occurrences before January 1st, 2010 or after September 1st, 2020; explicit mention of the total number of injuries or deaths being less than three; non-traumatic mass-casualty events; location of MCI solely outside of Rwanda; no mention of a specific event (e.g. descriptions of annual trends) or no mention of time and location. Articles were split by two study members (LV and MD) who completed eligibility screening, and data were extracted through RedCap (S2 Appendix).

### Data analysis

In cases where different articles described an MCI with the same trauma mechanism, province, and date (+/- one day), this was considered as the same event, unless there was information that indicated these were distinct MCIs. Articles describing the same MCI were merged and data was cleaned as following: in cases of discrepancies regarding the number of injuries, the largest number was chosen to avoid missing any data or on-site deaths; in case of discrepancies in terms of which districts were affected, the largest number of districts was chosen. Epidemiological patterns including the number of MCIs, number of injuries and on-site deaths (although not included in the MCI definition, considered as a marker of severity of an MCI), and temporal and geographical trends in MCI incidence, type, and number of injuries and deaths were analyzed using descriptive statistics and Fisher's exact test using Stata v16.0 (College Station, TX: StataCorp LP). P-values <0.05 were considered statistically significant.

## Results

The search strategy identified 3187 articles (3026 in English, 161 in French, and none in Kinyarwanda), of which 247 met inclusion criteria (S3 Appendix) and were included in the

study. The most common source was The New Times (95/247, 38.5%), which is Rwanda's leading newspaper. The 247 articles included in this study described 117 mass-casualty incidents, MCIs (S4 Appendix). For most trauma events, data on the number of injuries (n = 99, 84.6%), on-site deaths (n = 114, 97.4%) and the district(s) where the trauma occurred (n = 113, 96.6%) was provided. In 60.0% (n = 67) of the MCIs, data was available regarding which hospitals the patients were taken to: with the most common referral being to tertiary hospitals (n = 26, 38.8%), followed by a combination of tertiary, provincial, and/or district hospitals (n = 20, 29.9%), and district hospitals (n = 14, 20.9%). Of the 67 events with available information, patients injured during the same event were taken to more than one hospital in 43.3% of cases (n = 29). Demographic data were reported in 43 events (37.4%), of which 32 (27.4%) reported the name of ≥1 victim and 13 (11.3%) reported the age of ≥1 victim. Due to lack of consistent reporting, demographic data were excluded from further analysis.

## Characteristics of mass-casualty incidents

Of 117 MCIs, 73 (62.4%) were caused by road-traffic accidents, 23 (19.7%) by natural hazards, 20 (17.1%) acts of violence/terrorism, and 1 (0.09%) by a boat collision. The total incidence of MCIs, and the number of road-traffic accidents and acts of violence/terrorism-related events, decreased over time, although the number of natural hazard events increased, making it the most common mechanism of mass-trauma between 2017–2020 (Fig 1).

For road traffic accidents (n = 73), almost equal proportions were due to derailment of the vehicle off the road/overturned (n = 35, 47.9%) and collisions (n = 34, 46.6%). The vehicles most frequently involved were bus/cars (n = 58, 79.5%), trucks (n = 29, 39.7%), motorcycles (n = 5, 6.9%) and bicycles (n = 3, 4.1%). Pedestrians were involved in 11 accidents (15.1%).

For natural hazards (n = 23), landslides/flooding were the most common (n = 18, 78.3%), followed by earthquakes (n = 2, 8.7%), storms (n = 2, 8.7%), and lightning (n = 1, 4.3%). For acts of violence/terrorism(n = 20), explosion/detonation was the most common mechanism (n = 16, 80.0%), followed by firearms (n = 2, 10.0%), machete/knife (n = 1, 5.0%), and a combination of firearms and machete/knife (n = 1, 5.0%).

## Geographical distribution of events

33 (28.7%) MCIs took place in the Western province, 28 (24.3%) in the Southern province, 27 (23.5%) in Kigali, 25 (21.4%) in the Northern Province, 16 (13.7%) in the Eastern province, and one event where the province was not disclosed. Of these, 13 (11.1%) occurred over multiple provinces; all multiprovincial events were natural hazards. All but one of Rwanda's 30 districts (Nyamagabe, Southern province) had at least one MCI during the study period (Fig 2).

In Kigali, acts of violence/terrorism were the most common MCI, although reported as rare, isolated terrorist-related shootings and grenade blasts, where 75% occurred between 2011–2013, and only one act of violence/terrorism-related MCI occurred between 2017–2020. In the other regions, road-traffic accidents were most common, except for the Western province where road-traffic accidents and natural hazards were equally common (Table 1). The difference in proportions of trauma mechanisms, calculated for road-traffic accidents, natural hazards, and acts of violence/terrorism, was statistically significant for Kigali (p<0.000), the Western province (p<0.000) and the Southern province (p = 0.044), but not for the other provinces (Northern: p = 0.462; Eastern: p = 0.117).

## Injuries and deaths

The median number of injured persons per event was 11 (IQR 5–18), and the median number of on-site deaths was 2 (IQR 1–6). These numbers varied depending on the trauma

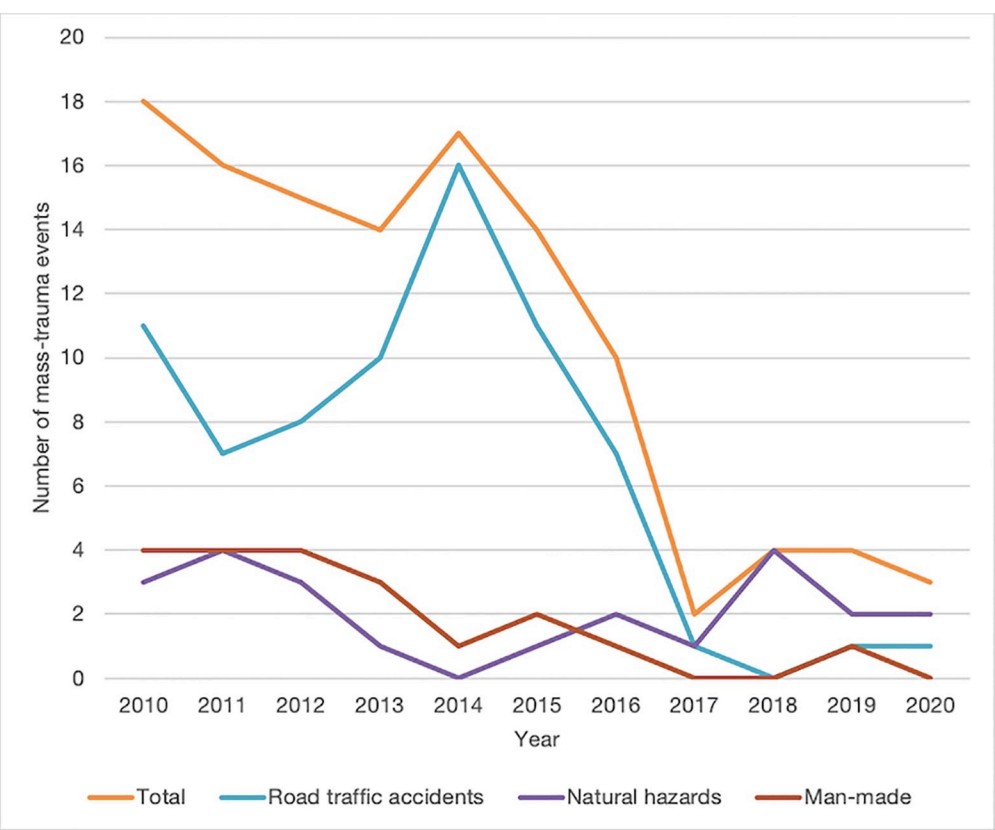

**Fig 1. The number of mass-casualty incidents plotted over time, reported by media.**

mechanism, with natural hazards having the highest median deaths (6 [IQR 2–18]), followed by a boat-accident (categorized as "other") (5 [IQR 5–5]), and road-traffic accidents (2.5 [IQR 1–6]). The boat-accident had the highest median injuries (37 [IQR 37–37]), followed by natural hazards (16.5 [IQR 6–22]), and road-traffic accidents (10.5 [IQR 5–15]). The number of deaths showed a downward trend from 2010–2017 but spiked in 2018 (Fig 3), when lightning struck a crowded church, injuring 140 persons.

## Discussion

The "systematic media review" is a novel methodology to assess mass-trauma epidemiology in absence of systematic data collection. Although commonly used in medicolegal research and used in one 2009 local case-series analysis of the Maryland health services [24], this is, to our knowledge, the first time that NexisUni is used for epidemiological research. Difficulties in developing global estimates of disease burden and epidemiological patterns are well-known, with most information being derived from UN data or academic institutions in the Global North [25]. For settings with limited resources and data collection, we propose the systematic media review as a feasible and cost-effective method with the potential to contribute to filling the global knowledge gap on MCIs, which in turn may help inform policy and clinical decision-making.

To our knowledge, there are no longitudinal studies describing the epidemiology of mass-trauma on a global scale. However, traumatic injury, measured by mortality and disability-adjusted life years, is decreasing in Rwanda and globally [26,27]. In this study, we similarly

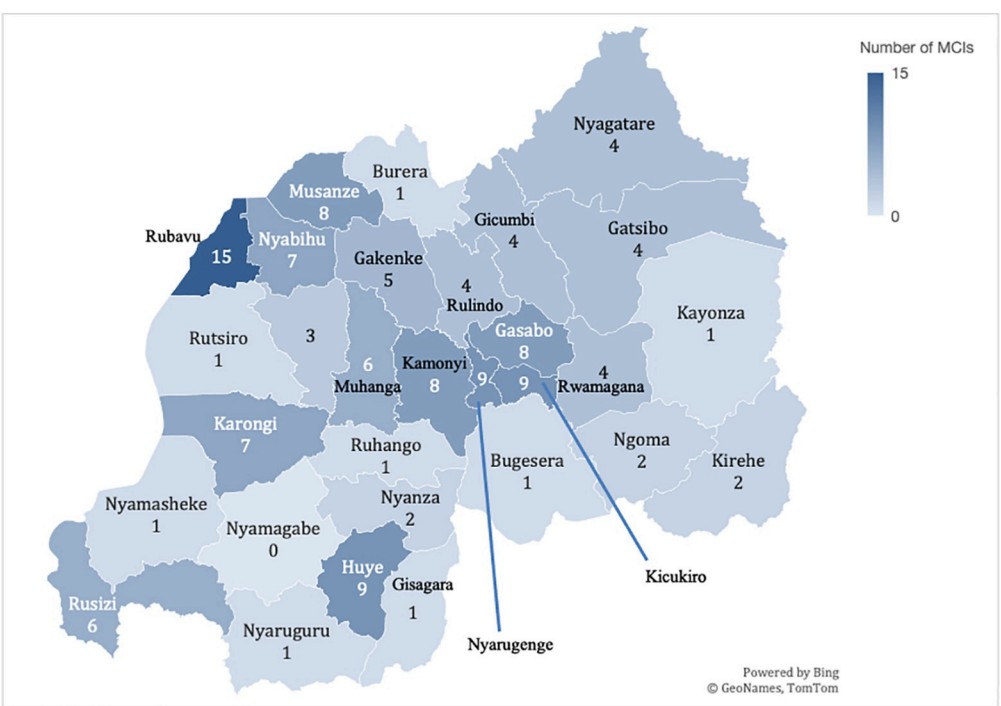

**Fig 2. Geographical distribution of mass-casualty incidents in Rwanda.** Map showing Rwanda's districts with the depth of color corresponding to the number of MCIs (mass-casualty incidents) reported by media between January 1st, 2010 and September 1st, 2020.

found that mass-trauma events and the associated injuries have decreased in Rwanda during the past 10 years. Yet, the number of deaths due to mass-trauma events did not improve during the study period, which may be explained by the relative increase in natural hazards that tend to be associated with high victim tolls [28,29]. Increased frequency and intensity of natural hazards is a known effect of climate change [30,31]. Countries with weak health infrastructure are especially vulnerable [32] and it is therefore imperative that developing trauma systems and national surgical/disaster plans also take into consideration changing meteorologic patterns [33,34]. According to recent data by the Ministry in Charge of Emergency Management, landslides and floods are the predominant types of natural hazards in Rwanda [14,15], when

**Table 1. Regional distribution of mass-casualty incidents by sub-type of incident.**

|  | All MCIs, n (%) | Causes of MCIs, n (%) | | | |
|---|---|---|---|---|---|
|  |  | **Road-traffic accidents** | **Natural hazards** | **Acts of violence/terrorism** | **Other (boat accident)** |
| **Total**[*] | 117 | 73 (62.4) | 23 (19.7) | 20 (17.1) | 1 (0.1) |
| **Kigali** | 27 | 12 (44.4) | 2 (7.4) | 13 (48.1) | 0 (0) |
| **Northern** | 25 | 15 (60.0) | 7 (28.0) | 3 (12.0) | 0 (0) |
| **Western** | 33 | 15 (45.5) | 15 (45.5) | 2 (6.1) | 1 (3.0) |
| **Southern** | 28 | 22 (78.6) | 5 (17.9) | 1 (3.6) | 0 (0) |
| **Eastern** | 16 | 9 (56.0) | 6 (37.5) | 1 (6.3) | 0 (0) |
| **Province not mentioned** | 1 | 0 (0) | 1 (100) | 0 (0) | 0 (0) |

Distribution of mass-casualty incidents (MCIs) reported by media, according to geographical province and trauma mechanism.

[*] As some events were multi-provincial the vertical sum may exceed the total number of events.

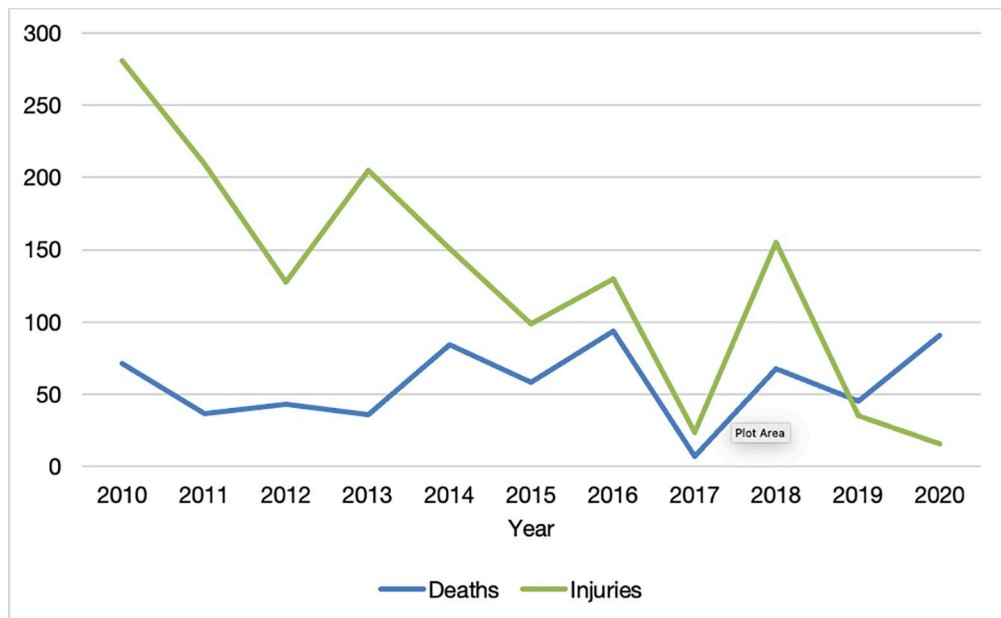

**Fig 3. The number of on-site deaths and injuries due to mass-trauma per year, as reported by media.**

excluding lightning which is not typically included in the four main categories of natural disasters: hydrological, meteorological, climatological, and geophysical disasters [35]. In our study, natural hazards, including lightning, was also the trauma mechanism with the largest number of on-site deaths, and the second highest median number of people injured, when excluding the single boat-accident categorized as "other". Although there is limited data on the direct impact of floods [36], floods lead to increased injury and mortality compared to landslides [37,38], which aligns with our findings. In addition to the consequences on morbidity and mortality, landslides and floods cause large socioeconomic costs, as houses and workplaces may be destroyed, but these aspects were outside the aims of this study [32].

Road-traffic accidents accounted for the bulk of mass-trauma in Rwanda until 2017, which resembles patterns reported by high-income countries [9,39] and the neighboring country Tanzania [40]. Globally, road-traffic accidents cause approximately 1.35 million deaths annually and between 20–50 million non-fatal injuries, with 93% of fatalities coming from LMICs [41]. The African Region counts 40% more road deaths per 100,000 people compared to other LMICs, and according to the World Health Organization, most injuries and deaths involve vulnerable road users, such as pedestrians, cyclists, and motorcyclists [41]. In our findings, car and bus passengers were most affected, and pedestrians and bicyclists were involved in 15.1% and 4.1% of MCIs, respectively, likely reflecting the fact that road-traffic accidents amounting to MCIs will be less likely to involve pedestrians compared to road-traffic accidents in general. This is lower than a study from Kigali where pedestrians were involved in 35% of accidents in 2012–2016 [42], although differences between Kigali City and national data may be explained by differences in urban-rural environments. The sharp decrease of road traffic accidents may be explained by the national road safety initiative, launched in 2004, where new laws were introduced to penalize excessive speed and drunk driving, and campaigns to raise awareness regarding road safety, and mandating speed governors on all public buses [43]. Consequently, casualties due to road traffic accidents decreased by 30%, and Rwanda has been championed as a successful example in improving road safety [43].

Our results showed a geographical skew of events, with natural hazards significantly more common in the Western province which is more mountainous and bordering lake Kivu, and road-traffic accidents significantly more frequent in the Southern province and Kigali. The increased vulnerability of the Western province matches the high-risk districts for natural disasters identified in the Rwandan National Contingency Plan on Floods and Landslides [15]. This geospatial pattern differs from e.g. the U.S. where the mass-trauma burden is predominantly concentrated in urban centers, likely due to larger proportions of road-traffic accidents and gun violence [9]. Our findings suggest that it could be appropriate in Rwanda to design sub-national strategies including prioritization of MCI protocols at provincial or district hospitals in rural areas, where over 80% of the population lives and where the most casualties caused by natural hazards occurred in the last years [44]. Additionally, referral guidelines for low-level health care facilities should be implemented, including bypass protocols to quickly enable transferring of urgent cases to higher-level facilities [45,46], to improve patient outcomes during MCIs [10]. However, the hilly topography and limited accessibility to certain areas [15], particularly in the rain season, are anecdotally known as complicating factors that delay transfers of victims of landslides and floods from the trauma site to medical centers. Future strategies to manage MCIs should therefore also consider including pre-hospital care in such strained circumstances.

Finally, we propose the "systematic media review" as a method to assess MCI epidemiology in the absence of databases, utilizing a systematic search strategy and a standardized, contextually adapted data collection form modified for the media review (S5 Appendix). We believe that this methodology has a great potential of filling a large data gap on mass-trauma epidemiology in settings lacking systematic data collection. With the level of detailed data appearing to be lower than data collection through clinical trauma registries, it would be ideal if this method could be seen as a complement to clinical data collection to ensure an in-depth understanding of the implications on the wider health system. Furthermore, in countries such as Rwanda, where incidence of trauma is high and access to care limited, the burden of trauma in patients who did not reach care is still unknown–a knowledge gap which may be addressed through the pre-hospital data collection in a systematic media review. However, with many LMICs lacking resources to develop and maintain trauma registries, the systematic media review could be a cost-effective and easy alternative, with the flexible methodology likely generalizable to different contexts although this should be assessed in future studies. Utilizing the systematic media review, we are introducing a new stakeholder to better understand mass-trauma epidemiology: journalists, who by profession are experts in data collection. This expertise could be further utilized by optimizing the strategy in collecting "pre-hospital data", which already occurs through news reporting, including epidemiologically relevant demographic factors, such as age or sex. In our study, national newspapers were the most common data sources, and most articles detailed in which district the trauma occurred, the number of on-site injuries, on-site deaths, and in many cases, to which hospitals the patients were transferred (60.0%). Training journalists in reporting trauma data in a systematic and standardized way could therefore be an innovative and cost-effective solution to overcome difficulties in collecting MCI data in LMICs.

## Limitations

This study has some limitations. Firstly, the term "injuries" can range from very mild injuries not needing healthcare to severe injuries leading to death, which makes comparisons between events difficult. Similarly, the threshold for what constitutes mass-casualty incidents in Rwanda has not been previously determined and therefore, our definition was arbitrary. A

previous study of surge capacity at provincial/tertiary hospitals in the neighboring country Tanzania used a fixed number of 10 for a road traffic accident to be considered a MCI [40]. Therefore, the threshold of three may be appropriate in rural areas of Rwanda, although it should possibly be adjusted upwards in urban areas.

Secondly, we don't know how many MCIs have not been reported and therefore can't state whether this method is exhaustive. By training journalists in MCI reporting and validating findings through other data sources, this knowledge gap could potentially be overbridged.

Thirdly, in cases where multiple articles described the same event, there were sometimes discrepancies in the number of people injured/deaths reported. This was particularly common for natural hazards, which, unlike road traffic accidents, are not always clearly limited in time and exact location. Although some demographic data on MCI victims were provided, the relative lack of details could make it difficult to use this method to track clinical outcomes.

## Conclusion

The systematic media review can be used to assess mass-trauma epidemiology in contexts where systematic data collection on MCIs is limited. In Rwanda, the number of MCIs has decreased, although landslides/floods are increasing, hindering a decrease in the number of deaths. MCI protocols in Rwanda should put an emphasis on rural areas, where most natural hazards occurred, and include modified referral protocols for critical patients. To improve the quality of the systematic media review, there is potential in training journalists in "mass-casualty reporting". Further studies will pair this novel method with clinical data collection to validate the method and to give a more granular understanding of the epidemiology of MCIs in Rwanda.

## Supporting information

**S1 Appendix. NexisUni search strategy.**
(DOCX)

**S2 Appendix. REDCap data extraction form.**
(DOCX)

**S3 Appendix. Included articles.**
(DOCX)

**S4 Appendix. All mass-trauma events identified in the media review.**
(DOCX)

**S5 Appendix. Proposed data collection form for systematic media reviews.**
(DOCX)

**S1 Raw data.**
(PDF)

**S1 Data.**
(XLSX)

## Acknowledgments

The map in Fig 2 is a Microsoft product screen shot reprinted with permission from Microsoft Corporation.

## Author Contributions

**Conceptualization:** Lotta Velin, Mbonyintwari Donatien, Andreas Wladis, Menelas Nkeshimana, Robert Riviello, Jean-Marie Uwitonze, Jean-Claude Byiringiro, Faustin Ntirenganya, Laura Pompermaier.

**Data curation:** Lotta Velin, Mbonyintwari Donatien.

**Formal analysis:** Lotta Velin, Mbonyintwari Donatien.

**Investigation:** Lotta Velin, Laura Pompermaier.

**Methodology:** Lotta Velin, Andreas Wladis, Menelas Nkeshimana, Robert Riviello, Jean-Marie Uwitonze, Jean-Claude Byiringiro, Faustin Ntirenganya, Laura Pompermaier.

**Project administration:** Lotta Velin.

**Resources:** Lotta Velin.

**Software:** Lotta Velin.

**Supervision:** Andreas Wladis, Menelas Nkeshimana, Robert Riviello, Laura Pompermaier.

**Validation:** Lotta Velin, Laura Pompermaier.

**Visualization:** Lotta Velin.

**Writing – original draft:** Lotta Velin, Mbonyintwari Donatien.

**Writing – review & editing:** Mbonyintwari Donatien, Andreas Wladis, Menelas Nkeshimana, Robert Riviello, Jean-Marie Uwitonze, Jean-Claude Byiringiro, Faustin Ntirenganya, Laura Pompermaier.

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
