## [Decision Letter · Decision Letter 0]

26 Jul 2021

PONE-D-21-19748

Systematic media review: a novel method to assess mass-trauma epidemiology in absence of databases - a pilot-study in Rwanda.

PLOS ONE

Dear Dr. Velin,

Thank you for submitting your manuscript to PLOS ONE. After careful consideration, we feel that it has merit but does not fully meet PLOS ONE’s publication criteria as it currently stands. Therefore, we invite you to submit a revised version of the manuscript that addresses the points raised during the review process.

Please consider all comments

We look forward to receiving your revised manuscript.

Kind regards,

Ahmed Mancy Mosa, Ph.D.

Academic Editor

PLOS ONE

Journal Requirements:

2. We note that Figure 2 in your submission contain map images which may be copyrighted. All PLOS content is published under the Creative Commons Attribution License (CC BY 4.0), which means that the manuscript, images, and Supporting Information files will be freely available online, and any third party is permitted to access, download, copy, distribute, and use these materials in any way, even commercially, with proper attribution. For these reasons, we cannot publish previously copyrighted maps or satellite images created using proprietary data, such as Google software (Google Maps, Street View, and Earth). For more information, see our copyright guidelines: http://journals.plos.org/plosone/s/licenses-and-copyright.

Reviewers' comments:

Reviewer's Responses to Questions

**Comments to the Author**

1. Is the manuscript technically sound, and do the data support the conclusions?

Reviewer #1: Yes

Reviewer #2: Partly

2. Has the statistical analysis been performed appropriately and rigorously? 

Reviewer #1: Yes

Reviewer #2: No

3. Have the authors made all data underlying the findings in their manuscript fully available?

Reviewer #1: Yes

Reviewer #2: Yes

4. Is the manuscript presented in an intelligible fashion and written in standard English?

Reviewer #1: Yes

Reviewer #2: Yes

5. Review Comments to the Author

Reviewer #1: Comment PLOS One - PONE-D-21-19748

Title – Systematic media review: a novel method to assess mass-trauma epidemiology in absence of databases - a pilot-study in Rwanda

Abstract – Structured and well written

Background –

Detailed information on statement of problem as well as rational for the study clearly presented.

Methods

Well described in details

Result

Well written in details with relevant tables and figures

Page 13, line 168 – 170: Is this multiple response? The different types of transport / vehicle added together is more than 100%. This could not be verified on any of the tables

Discussion

Well discussed with study limitations provided.

Conclusion –

Clearly written with appropriate recommendation.

Reviewer #2: Systematic media review a novel method to assess mass-trauma epidemiology in absence of database-a pilot study in Rwanda

This is an important study that could provide input for health planners and policy makers in saving the lives of many in Rwanda and as well exemplary for other LMICs.

• Much emphasis is given to the problem of MCI in all over the world than the current situation in the local setting about recording of the MCI. To consider the novelty of the proposed system, we need to know the current situation of recording/registration of MCI in Rwanda. The novelty of the method is less emphasized in the whole presentation of the article. Injury registry has been implemented [lines 92-93, page 4] so where is the novelty of your method? Do you mean this approach will provide information broader in scope as previous data were limited to certain types of disasters [line 93]? It seems that nothing is special here in Rwanda to implement the current study with novel approach. Please first provide the big picture of the problem i.e., the legacy system that the novel approach could improve.

• The rationale presented [lines 96-97, page 4] to estimate the burden of MCI in limited resource setting is important. Such report on descriptive type of epidemiology [Person, Place, and Time] is the main component expected from the analysis. The trend in the traumatic injury [Time] is reported [Fig. 1], the geographic distribution [Place] where all the incidents happened during the study period is also well presented [Fig. 2]. However, it seems that the current analysis missed a big component of the demographic structure [Person] of the traumatic injury report to estimate the burden of MCI. Without personal identifier, the age distribution and gender mix of MCI should have been reported to estimate the burden. I think none of the search strategy could not capture such information [Appendix 1]. Age and gender should be the component as keyword in your search strategy to have the demographic distribution. Since such keywords were not used your result could not capture this information and it was reported as if age and gender were not reported less. Demographic data were reported in 37.4% [lines 156-158, page 6], but such inconsistency is a result of the search strategy.

Methods

• The working definition given to MCI is good, but it focused on the frequency of event. What parameters you considered to arrive such arbitrary number should be explained [e.g. hospitals on average in Rwanda can manage at the same time]. Further, your definition should include the nature of injuries. Does your definition of MCI work for all-natural calamities and human caused injury incidents contrary to the definition you mentioned [lines 59-60, page 3]?

• Is death included in your definition of MCI and in the reported traumatic injury? Your result includes death pattern as MCI [line 138-139, page 6]

• Please add a few detailed descriptions to the features of the database you used (NexisUni).

• For traumatic injury happened in neighboring/adjacent districts outside Rwanda, for example, in a district in Tanzania, but cases were referred to health facilities in one of the districts in Rwanda making the healthcare system overburdened. How do you verify such incidents to include/not in the analysis?

• Your search strategy indicates the input data sources used are electronic media. If the MCI happened and document in a non-electronic media, for example, paper-based news report, how such traumatic injury reports were managed not to miss incidents in your analysis?

• In the data analysis subsection, it is reported that Fisher’s exact test is used, but the result presentation did not show any findings related to such analysis.

Results

• Any mechanism applied to verify for cases not to be counted twice? In the place where the incident happened and if they referred into the next higher level in the health care system and got news attention there will be a possibility of multiple count of cases. [lines 154-156, page 6].

• Figure 2 needs revisions on the colour resolution applied. The legend showing the colour depth ranges from 0-15 and 0 should be white as there is no value. But that is not the case in the figure. It is good to add values in the picture for each region together with the label name for each district.

Discussion

• Mortality in mass-trauma events did not improve during the study period [lines 220-221, page 9]. Where this result and conclusion come from as the definition did not include death rather the number of injuries was a prioritized concept?

• Please reflect more on the novelty of the method together with its possible benefits and advantages the case at hand over the already existed system.

• In one of the limitations of the study, the current study finding is compared with respective ministry reports. Such analysis is not evidenced in the result section and no attempt was made to verify/validate the current study estimate against state report. Why it appeared out of the blue in the limitation?

• it is suggested that MCI protocols in Rwanda should put an emphasis on rural areas. But for international readers to take this as a key message from the current study, please show the urban rural disparity in terms of MCI in the geographic distribution analysis.

6. PLOS authors have the option to publish the peer review history of their article (what does this mean?). If published, this will include your full peer review and any attached files.

Reviewer #1: **Yes: **Prof. Tanimola Akande

Reviewer #2: No

---

## [Author Response · Author response to Decision Letter 0]

30 Aug 2021

We want to thank you for taking your time to review our text in-depth and for the valid and constructive feedback provided. We have addressed your comments in our separate "response to reviewers" and made changes accordingly, which we truly believe will make this paper better. If anything is unclear, please do not hesitate to reach out to us for further questions or clarification.

---

## [Decision Letter · Decision Letter 1]

28 Sep 2021

Systematic media review: a novel method to assess mass-trauma epidemiology in absence of databases - a pilot-study in Rwanda.

PONE-D-21-19748R1

Dear Dr. Velin,

We’re pleased to inform you that your manuscript has been judged scientifically suitable for publication and will be formally accepted for publication once it meets all outstanding technical requirements.

Kind regards,

Ahmed Mancy Mosa, Ph.D.

Academic Editor

PLOS ONE

Additional Editor Comments (optional):

Reviewers' comments:

Reviewer's Responses to Questions

**Comments to the Author**

1. If the authors have adequately addressed your comments raised in a previous round of review and you feel that this manuscript is now acceptable for publication, you may indicate that here to bypass the “Comments to the Author” section, enter your conflict of interest statement in the “Confidential to Editor” section, and submit your "Accept" recommendation.

Reviewer #1: All comments have been addressed

Reviewer #2: All comments have been addressed

2. Is the manuscript technically sound, and do the data support the conclusions?

Reviewer #1: Yes

Reviewer #2: Partly

3. Has the statistical analysis been performed appropriately and rigorously? 

Reviewer #1: Yes

Reviewer #2: Yes

4. Have the authors made all data underlying the findings in their manuscript fully available?

Reviewer #1: Yes

Reviewer #2: Yes

5. Is the manuscript presented in an intelligible fashion and written in standard English?

Reviewer #1: Yes

Reviewer #2: Yes

6. Review Comments to the Author

Reviewer #1: (No Response)

Reviewer #2: (No Response)

7. PLOS authors have the option to publish the peer review history of their article (what does this mean?). If published, this will include your full peer review and any attached files.

Reviewer #1: **Yes: **Prof. Tanimola Akande

Reviewer #2: No

---

## [Editor Report · Acceptance letter]

5 Oct 2021

PONE-D-21-19748R1 

Systematic media review: a novel method to assess mass-trauma epidemiology in absence of databases - a pilot-study in Rwanda. 

Dear Dr. Velin:

I'm pleased to inform you that your manuscript has been deemed suitable for publication in PLOS ONE. Congratulations! Your manuscript is now with our production department. 

Kind regards, 

on behalf of

Dr. Ahmed Mancy Mosa 

Academic Editor

PLOS ONE